# Scalable trapping of single nanosized extracellular vesicles using plasmonics

Chuchuan Hong[1,2] & Justus C. Ndukaife [1,2,3] ✉

Heterogeneous nanoscale extracellular vesicles (EVs) are of significant interest for disease detection, monitoring, and therapeutics. However, trapping these nano-sized EVs using optical tweezers has been challenging due to their small size. Plasmon-enhanced optical trapping offers a solution. Nevertheless, existing plasmonic tweezers have limited throughput and can take tens of minutes for trapping for low particle concentrations. Here, we present an innovative approach called geometry-induced electrohydrodynamic tweezers (GET) that overcomes these limitations. GET generates multiple electro-hydrodynamic potentials, allowing parallel transport and trapping of single EVs within seconds. By integrating nanoscale plasmonic cavities at the center of each GET trap, single EVs can be placed near plasmonic cavities, enabling instant plasmon-enhanced optical trapping upon laser illumination without detrimental heating effects. These non-invasive scalable hybrid nanotweezers open new horizons for high-throughput tether-free plasmon-enhanced single EV trapping and spectroscopy. Other potential areas of impact include nano-plastics characterization, and scalable hybrid integration for quantum photonics.

Once thought of as a means for cells to expel wastes, in recent years, nanosized extracellular vesicles[1] have generated substantial scientific interest because they contain important biological molecules, including proteins, lipids, and nucleic acids, and serve as a means for cells to communicate with neighboring or distant cells[2]. EVs are heterogeneous in size, biogenesis, and molecular composition[3,4]. They comprise of exosomes derived from multivesicular bodies that mature and attach to the cell membrane for release, ectosomes derived from an outward budding of the cell membranes, exomeres discovered in 2018[5], and the recently discovered supermeres[6], which are only 25 nm in size. The biogenesis mechanisms of both exomeres and supermeres are currently unknown[5]. Given their heterogeneity, one of the key challenges limiting an enhanced understanding of EVs is the lack of suitable tools for trapping and analyzing them at the individual vesicle level[7]. Developing such a capability would provide the means to study a large population of EVs to understand their

heterogeneity in size and molecular composition at the single particle level and drive translational biomedical applications.

Optical tweezers[8] recently recognized with a 2018 Physics Nobel prize have emerged as a powerful tool for the manipulation of microscopic biological objects such as cells, colloidal assembly, and single particle spectroscopy[9]. In optical tweezers, a tightly focused laser beam is used to generate strong gradient forces on microscopic objects to stably trap them at the laser focus. Unfortunately, due to the diffraction limit of light, the low-power stable trapping of nanosized objects such as nanosized EVs has been met with challenges. Optical tweezer-based Raman spectroscopy requires high laser power (~100 mW or higher) to stably trap EVs[10–13], which sometimes leads to the explosion of EVs that interferes with subsequent measurements[10]. Lowering the laser power brings about multiple issues, including insufficient trapping stability so that the particle escapes from the trap before the completion of the signal collection and restriction on the

[1]Department of Electrical and Computer Engineering, Vanderbilt University, Nashville, TN, USA. [2]Vanderbilt Institute of Nanoscale Science and Engineering, Vanderbilt University, Nashville, TN, USA. [3]Department of Mechanical Engineering, Vanderbilt University, Nashville, TN, USA. ✉e-mail: justus.ndukaife@vanderbilt.edu

size of vesicles that can be trapped[13]. Furthermore, optical tweezers do not guarantee single EV trapping and often result in the collection of multiple EVs within the diffraction-limited laser spot. Furthermore, the loading of EVs into the optical trap is a slow process that can take several minutes, which will adversely impact the analysis throughput[10]. Towards addressing the limited trapping stability of conventional optical tweezers, near-field optical nanotweezers based on plasmonic cavities[14–18] have been investigated. The plasmonic cavities efficiently couple to propagating light to generate an enhanced and spatially confined electromagnetic field well below the diffraction limit[19]. Early developments of plasmonic tweezers rely on Brownian diffusion to load the trap, which is a slow, nondeterministic, and time-consuming process that makes plasmonic tweezers impractical for low particle concentration solutions[20]. Prior strategies using convection flows and thermophoresis to load the trap suffer from particle aggregation and are not suitable for single particle trapping[21,22]. The development of electrothermoplasmonic (ETP) tweezers[23–25], which harness plasmonic heating with an applied alternating current electric field to induce electrothermoplasmonic flow for transport of particles towards the plasmonic cavity, enabled rapid trapping of particles within seconds at an illuminated plasmonic nanostructure. However, the ETP approach requires plasmonic heating to transport particles to the electromagnetic hotspot, which is also the location of the thermal hotspot, and thus poses photothermal damage to delicate biological specimens. If the photothermal heating[26] is dissipated such as by using a high thermal conductivity substrate as a heat sink[27], the local plasmon-induced temperature rise becomes absent, which comes with the disadvantage that electrothermoplasmonic flow can no longer be induced to enable particle transport to a trapping site. No approach reported to date is capable of achieving fast transport of single nanosized objects for stable trapping at a plasmonic hotspot, while ensuring that photothermal heating is eliminated at the plasmonic hotspot. The lack of such capability presents a barrier to harnessing plasmonic optical nanotweezers for the analysis of heterogeneous populations of nanosized biological objects like EVs. We report here our geometry-induced electrohydrodynamic tweezer (GET) that addresses the aforementioned challenges and meets the requirements for an ideal nanomanipulator for scalable single EV trapping. GET enables the massive parallel trapping of single nanosized objects, such as nanosized EVs within seconds near plasmonic hotspots without photothermal heating at the trapping site, thus ensuring that the trapping site is always a location of low-temperature rise. GET is also scalable and the number of single-particle GET traps is dependent on the number of trap sites defined on-chip via lithographic fabrication and can range from hundreds, thousands, or millions as desired. By integrating individual plasmonic cavities at the center of each GET traps, the electrohydrodynamic potential places particles in parallel within seconds in the vicinity of the plasmonic cavities to facilitate instantaneous near-field plasmon-enhanced optical trapping when any GET trap is illuminated with a focused laser without any detrimental photothermal heating. Our finding represents a significant development in the nanoscale trapping and optical nanomanipulation field by addressing the important task of fast parallelized trapping of single nanosized vesicles and particles within seconds, instantaneous plasmonic trapping with single particle resolution without the risk of photothermal damage, thus paving the way for high-throughput plasmon-enhanced single particle spectroscopies that are of interest in multiple areas[28].

## Results

### Working principle of GET
The GET platform is made up of a finite array of plasmonic nanoholes arranged in a circular geometry with an inner void region (Fig. 1a). This circular array of plasmonic nanoholes with a central void region performs two functions that include generating an electrohydrodynamic

potential for trapping single nanosized EVs at the center of each void region and exciting surface plasmon waves to outcouple and beam emitted photons from fluorescently labeled particles trapped at the center of the void regions as described in the following. First, to understand how the electrohydrodynamic potentials are generated in GET, we consider a square array of nanoholes on a gold film substrate as shown in Fig. 1b. If an electric field is applied perpendicular to the patterned gold substrate, the gold nanohole array distorts the a.c electric field lines giving rise to normal and tangential a.c. electric fields. The tangential component of the a.c. electric field exerts Coulombic forces on the diffuse charges in the electrical double layer induced at the interface between the nanohole array and the fluid, giving rise to an a.c. electro-osmotic motion[29,30] of the fluid that transports the suspended particles radially outwards away from the edge of the nanohole array, as depicted in Fig. 1b. If three additional nanohole array patterns are placed adjacent to the original nanohole array to form a central void region, as depicted in Fig. 1c, the radially outward a.c. electroosmotic flow will oppose each other in different directions creating a stagnation zone at the center. If the geometry of the nanohole array is transitioned from a square void region into a circular void region, opposing a.c. electro-osmotic flows will also form a stagnation zone at the center of the circular void region of the array of plasmonic nanoholes, as illustrated in Fig. 1d and Fig. 1a. This defines the position of the electrohydrodynamic potential minimum where a single particle can be readily trapped in GET. The localization of the single trapped EVs in the out-of-plane direction comes from the particle-surface interaction force that arises from the interaction between the double layer charge on the particle and its image charge in the conduction plane[31]. The number of GET traps that can be generated on-chip is scalable and is only limited by the number of lithographically defined nanohole array patterns with void regions fabricated on-chip thus enabling parallelized single nanoparticle trapping across multiple trapping sites on-chip. Besides generating an electrohydrodynamic trapping potential, the nanohole array also performs the function of enhancing the outcoupling of fluorescence emission from trapped single fluorescently-labeled EVs, as depicted in Fig. 1e. For this purpose, the nanohole array closest to the void region is fashioned to have a radial periodicity. The periodicity of the nanohole array is then tuned to excite surface plasmon waves that efficiently outcouples light within the emission wavelength of a given fluorophore and beams the emitted light into the collection optics (details in Supplementary Discussion III).

We have employed COMSOL Multiphysics and commercial finite-domain time-difference solver (Lumerical FDTD) to model the electrohydrodynamic trapping in GET as well as for the optimization of the geometry of the nanohole array to achieve enhanced imaging of trapped fluorescent EVs, respectively, as described in the Supplementary Discussion I and III respectively. The simulated a.c. electro-osmotic flows from COMSOL Multiphysics are presented in Fig. 1f.

### Parallel single-particle resolution trapping
The experimental demonstration of parallel trapping of single nanosized EVs and dielectric nanoparticles using GET was carried out using a circular array of gold nanoholes with a diameter of 160 nm and a thickness of 120 nm arranged to have a void region. The patterned gold nanohole array metasurface is on either a glass or sapphire substrate. The diameter of the void region ranges from 4 µm to 25 µm. The fabrication was performed using a template stripping approach[32] (see Methods section). The SEM image of the fabricated sample is shown in Fig. 1g. Experimental demonstration of trapping was performed using solutions of fluorescently labelled CD 63+ EVs (see Methods section) with a size range of 30 nm to 150 nm confirmed by nanoparticles tracking analysis, as well as 100 nm diameter polystyrene beads. The particle solution was diluted to concentrations ranging from $10^5$ to $10^9$ particles/ml. Initial experiments to demonstrate parallel trapping and

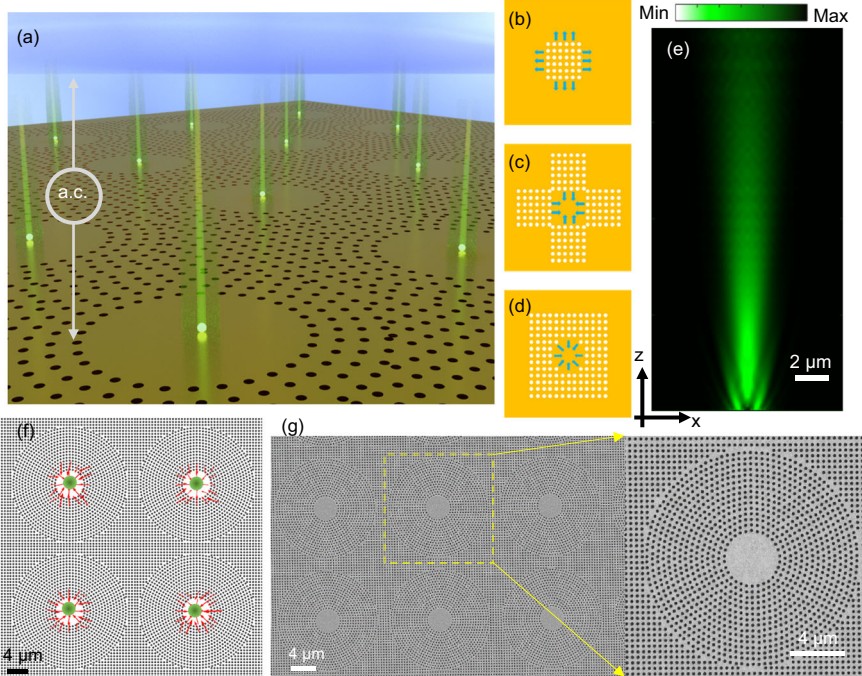

**Fig. 1 | Illustration and theoretical analysis of the GET system. a** Illustration of the operating mechanism of the GET system. The tangential a.c. field induces electro-osmotic flow that is radially outward. By harnessing a circular geometry with a void region, the radially outward a.c. electro-osmotic flow creates a stagnation zone at the center of the void region where trapping takes place. **b** A square-lattice nanohole array generates a.c. electro-osmotic flow outwards. **c** Four square lattice arrays create a.c. electro-osmotic flows converging to the center. **d** A radial-lattice nanohole array generates a.c. electro-osmotic flows converging to the center of the void region. **b**–**d** illustrate the evolution from a square-lattice nanohole array into a radial-lattice nanohole array. **e** Radiation energy flow for a dipole

fluorescence emitter placed at the center of the void region showing the ability to harness the GET trap to also beam emitted photons from trapped particles. **f** COMSOL simulation of the radial electro-osmotic flow showing that the geometry of the void region results in opposing electro-osmotic flow that forms a stagnation zone at the center. Particle trapping occurs at the center of the void region where the flow vectors converge. The particle trapping position is highlighted with green dots, **g** SEM image of the plasmonic metasurface array with void regions, and a zoomed-in version of an individual GET trap. Each void region represents a GET trap and can be readily scaled from hundreds to thousands or millions as desired.

for characterizing the speed of particle trapping in GET were performed using 100 nm diameter fluorescently-labeled polystyrene beads. After introducing the nanoscale polystyrene beads into the microfluidic channel containing the GET chip, an a.c. electric field of 83,333 V/m was applied across the microfluidic channel containing the GET chip. The frequency was tuned from 10 kHz to 3.5 kHz. This results in the fast parallelized transport of single particles towards each independent GET trap. Within three seconds, single nanosized polystyrene beads were shown to have been trapped in parallel at the center of each void region of the nanohole arrays that define the stagnation zones, as depicted in Fig. 2a (Supplementary Movie 1). These results depict that GET can rapidly transport and trap single nanosized particles in parallel within seconds across the chip without relying on slow Brownian diffusion. This rapid loading and trapping of nanosized particles in parallel are achieved for all the particle concentrations we examined ($10^5$–$10^9$ particles/ml). Details of the results are discussed in Supplementary Discussion IV and Supplementary Movie 7.

We have performed detailed experiments to determine the conditions and optimal design of GET traps for ensuring self-limiting single particle resolution trapping in GET. To achieve self-limiting single particle resolution trapping, we harness the interplay between the in-plane drag force from the a.c. electro-osmotic flows and the dipole-dipole repulsion force between particles. Our result (Fig. 2c) shows that using a void region of 4 μm diameter for an a.c. frequency above 3.5 kHz, the dipole-dipole repulsion force begins to overcome the drag force from a.c. electro-osmotic flow. Figure 2b (Supplementary Movie 2) shows the trapping of two particles in one GET site using an a.c. frequency of 2.5 kHz. When the a.c. frequency is increased to

3.5 kHz, the dipole-dipole repulsion force results in the expulsion of the second particle from the trap to ensure that only one particle occupies the GET well. Detailed calculation of the interplay between the drag force from a.c. electro-osmotic flow and dipole-dipole repulsion force is provided in Supplementary Discussion V.

The geometry of the nanohole array can also be optimized to boost the imaging of trapped particles. As depicted in Supplementary Discussion III and Supplementary Fig. 6, the optimal periodicity of the nanohole array boosts the fluorescence emission 3.1-fold relative to the unoptimized system by beaming the photons upward for efficient collection.

Figure 3a (Supplementary Movie 3) shows an image of trapped single EVs of varying sizes all trapped in parallel across the chip using an a.c. field of 3.5 kHz. Figure 3b shows the scatter plot of the trapped EV displacement. It is evident that the EV at a given trap experiences similar trapping stability along the x and y directions. This is expected since the GET traps have a circular geometry and hence are symmetric. We also note that the GET trap exhibits higher trapping stability at a lower a.c. frequency of 2 kHz. We attribute this to the fact that the particle-surface interaction force is stronger as the a.c. field frequency reduces since the double layer charges have sufficient time to be polarized by the applied a.c. field. The trapping stability is also dependent on the EV size with the larger EVs (95 nm) exhibiting a higher trapping stability than the smaller EVs (44 nm) for a given frequency as depicted in Fig. 3b. Due to the heterogeneous size distribution of EVs, we also performed correlative fluorescence and electron microscopy imaging to verify the diameter of the trapped EVs and correlate with the trapping stabilities. After trapping the single EVs in parallel, a low-frequency a.c. electric field with a frequency of 100 Hz

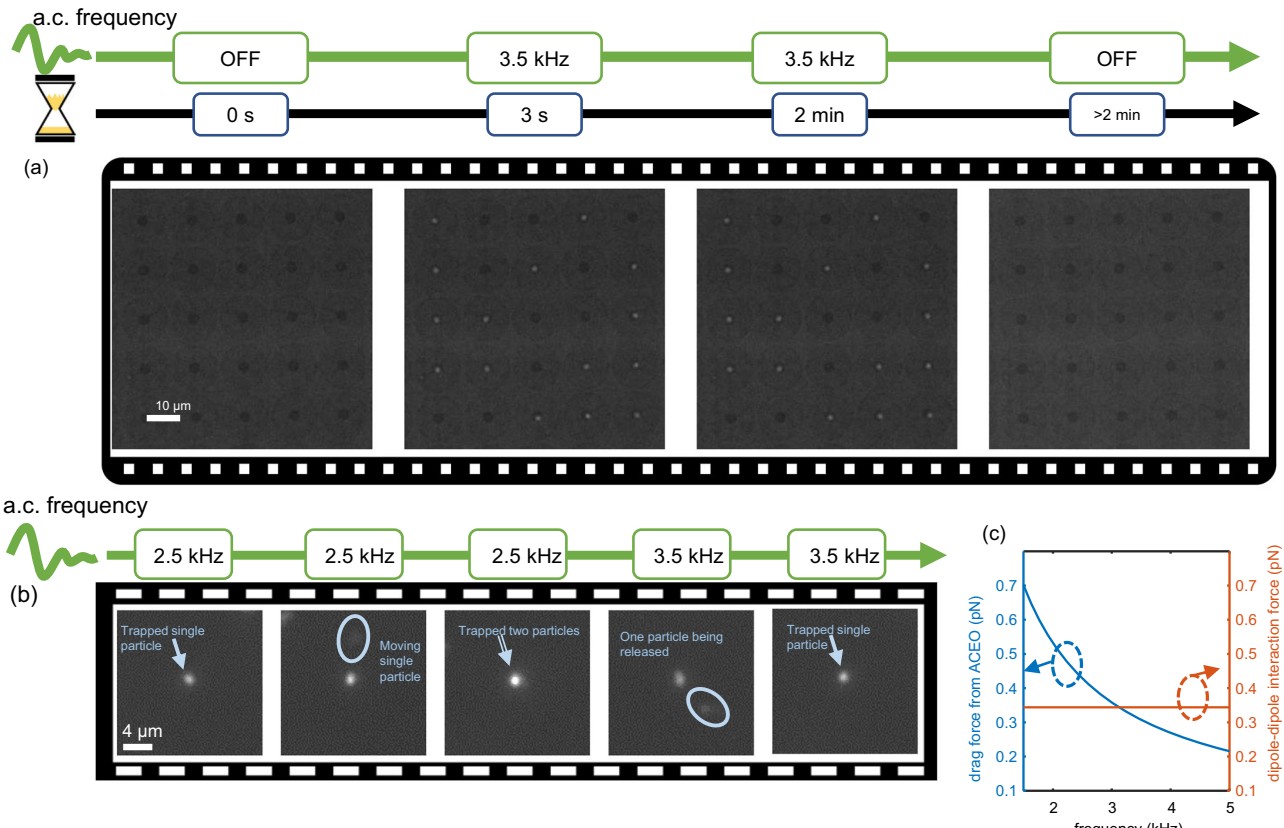

**Fig. 2 | Demonstration of parallel transport, single particle resolution trapping and release of a single 100 nm size polystyrene beads. a** A frame-by-frame sequence of the rapid-onset (3 seconds) and long-lasting (>2 min) stable parallel trapping when an a.c. field of 3.5 kHz is applied. The particles are trapped at the center of each GET trap. The trapped particles are released when the a.c electric field is turned off. **b** Demonstration of self-limiting single particle resolution trapping feature of GET. For an a.c. field of 2.5 kHz, a GET trap can take more than one particle. Two particles were shown to be trapped. Subsequently, as the frequency is increased to 3.5 kHz, the second particle is released from the trap due to the interplay between dipole-dipole repulsion force and radial a.c. electro-osmotic flow. **c** Comparison between dipole-dipole repulsion force and drag force from a.c. electro-osmotic flow. For a.c. frequencies above 3.5 kHz, the dipole-dipole repulsion force overcomes the electro-osmotic flow drag force.

was applied to immobilize the trapped EVs in position. The low frequency a.c. field results in an electrophoretic force that presses the EV to the surface so that van der Waal's force permanently holds them in place. The SEM images in the inset of Fig. 3b depict that the trapped EVs range in size from 44 nm to 95 nm, which corresponds to the size range of small EVs (30 nm to 120 nm). More discussion on the particle size trapped by the GET system is provided in Supplementary Discussion VI, and Supplementary Movie 8 and 9.

## Scalable plasmonic trapping of EVs with GET

Up to now, we have demonstrated the use of the electrohydrodynamic potential in GET for parallel transport, trapping and enhanced imaging of single nanosized EVs at the center of the void regions of the plasmonic nanohole array. The following section focuses on how the electrohydrodynamic potential can be superimposed with a plasmon-enhanced optical trapping potential to facilitate both parallel electrohydrodynamic trapping and plasmon-enhanced optical trapping on demand upon laser illumination without any detrimental heating effect. This feature is possible because in GET a single nanosized vesicle is readily trapped at the center of the void regions of the plasmonic nanohole array. Thus, if a plasmonic cavity, i.e. a nanoantenna is placed at the center of the void region, the electrohydrodynamic potential will become superimposed with the plasmon-enhanced optical trapping potential upon laser illumination of a given plasmonic cavity in a GET site. For experimental demonstration, we have introduced a plasmonic cavity comprising of a double nanohole

aperture in a metal film within a GET trap as depicted in Fig. 4a and b. To dissipate plasmonic heating and mitigate the laser-induced photothermal heating effect, the substrate on which the gold nanohole array is placed on is chosen as sapphire. The sapphire substrate has a higher thermal conductivity of 25.2 W/m · K in comparison to glass (1.38 W/m · K), and hence serves as a heat sink to dissipate excess heat from plasmon-enhanced absorption at the plasmonic hotspot of the double nanohole aperture plasmonic cavity. Figure 4c shows that the local temperature rise for an incident intensity of $3.2 \times 10^9$ W/m² (laser power of 6.3 mW) is only 0.32 K. This negligible temperature rise not only prevents trapped EVs from heating-induced damage, but also primarily suppresses the generation of thermal-related effects that destabilize the trapping, such as positive thermophoresis or convection. Furthermore, the low temperature rise achievable with a sapphire substrate suppresses ETP flow from destabilizing the particles trapped at nearby GET sites when another GET site is illuminated. Figure 4d shows that the plasmon-enhanced optical trapping potential on a 100 nm EV reaches $1.8 \, k_bT$, which is sufficient to stably trap the particle under the low temperature rise of 0.32 K. The recorded video is provided in Supplementary Movie 4. Initially, the a.c. field is turned ON and set to a frequency of 3.5 kHz to enable rapid parallel trapping of single EVs. The single EVs are positioned at the center of the void region, near the plasmonic double nanohole aperture and are held by the electrohydrodynamic potential as depicted in the first frame of Fig. 4e. At this point the GET system with centralized plasmonic cavities provides absolute freedom to select one of the following options:

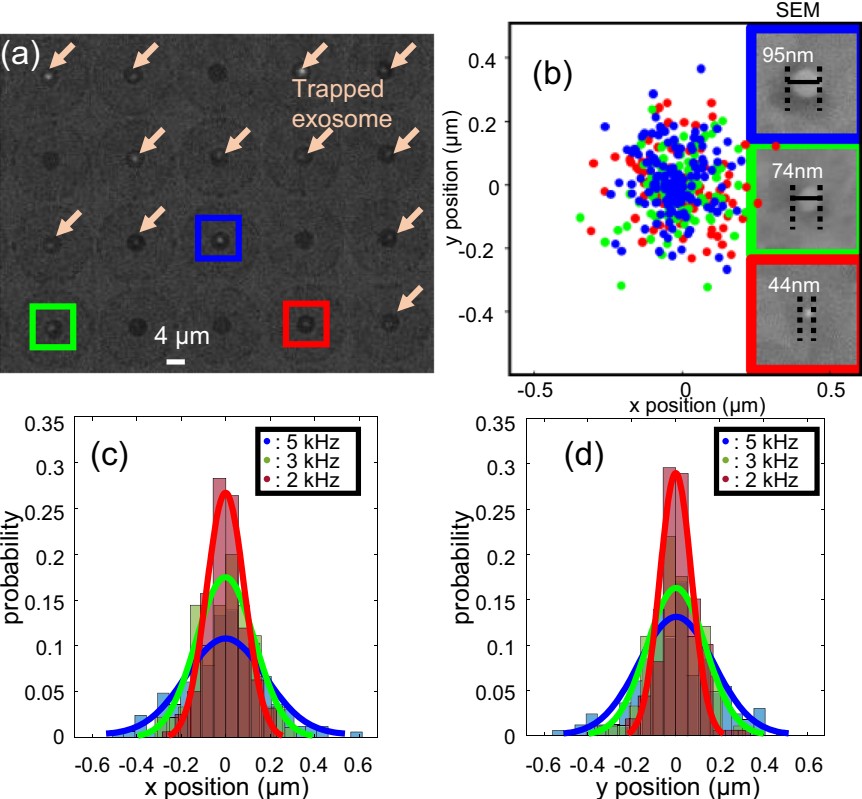

**Fig. 3 | Fluorescence optical image showing the parallel trapping of single nanosized EVs. a** This parallel trapping occurs across the multiple trapping sites defined on-chip. Arrows show the trapped EVs. **b** Scatter plots showing the trapping stabilities of the trapped EVs indicated in blue, green, and red boxes in (**a**) for an applied a.c. frequency of 3.5 kHz. The insets show the SEM images of the trapped EVs enclosed in blue, green, and red boxes, depicting that the sizes range between 44 nm and 95 nm. **c** and **d** Histograms of lateral displacement along the x and y directions on trapped small EV (44 nm), respectively. The curves show the Gaussian-shape fitted curve for clearer illustration.

(1) illuminating on a GET site containing a plasmonic cavity with a laser to precisely position the nanosized EV to the illuminated plasmonic hotspot using plasmon-enhanced optical force, while the other EVs at other GET sites are held by the electrohydrodynamic potentials; (2) moving the laser spot or stage to another GET trap to position another EV to the plasmonic hotspot with plasmonic-enhanced optical trapping; (3) turning OFF the a.c. field and keeping the laser ON to keep a particle plasmonically-trapped with plasmon-enhanced optical force; (4) releasing the trapped EVs by turning OFF both the laser and a.c. field; (5) immobilizing the particle at the plasmonic cavity by temporarily switching the a.c. field with a d.c. field or low frequency a.c. field below 100 Hz (Fig. 5b and Supplementary Movie 6). The first and second options provide the means to harness the GET system for both rapid high stability near-field optical trapping at plasmonic hotspots and plasmon-enhanced spectroscopies such as surface enhanced fluorescence and surface enhanced Raman spectroscopy of individual EVs. The last option terminates with a zero-power trapping that requires neither the a.c. field nor laser illumination, and thus permits additional off-chip analysis. The SEM image of an EV positioned precisely at the plasmonic cavity is depicted in Fig. 5c. Dynamic relocation of a trapped EV from one plasmonic hotspot to the next can also be achieved in the GET system by temporarily illuminating the nanohole array outside the void region with a laser power (25 mW) that is four times greater than the power required for plasmon-enhanced trapping (6.3 mW) as shown in Fig. 5a and Supplementary Movie 5. Using a higher laser power at the nanohole array increases the temperature rise from 0.32 K to up to 8.44 K as depicted in Supplementary Fig. 3 due to the increased power and the collective heating effect from the nanohole array. In the presence of the a.c. field, this slight temperature

rise in the nanohole array region induces a radially inward ETP flow (Supplementary Fig. 3) that releases a nearby EV and transports it towards another GET site. Once the laser is turned OFF or lowered, the ETP flow disappears, while the in-plane a.c. electro-osmotic flow acts to deliver the particle to the center of the nearest GET trap as depicted in the last frame of Fig. 5a. After the dynamic relocation, the laser power is reduced to 6.3 mW, which is sufficient for plasmon-enhanced trapping without any ETP flow. We note that during these processes of dynamic relocation, the EV never directly interacts with the laser beam focus, thus precluding any possibility for photothermal damage. Also, a long-lasting trapping of particle using the plasmonic cavity for 25 min does not induce damage to the trapped particle, as shown in Supplementary Movie 13. Details of the flow field simulation for the dynamic manipulation of a trapped EV from one plasmonic cavity to the next in GET are discussed in Supplementary Discussion II. We conducted further experiments to compare the effectiveness of plasmonic trapping with and without the GET system. Our findings indicate that the GET system is capable of trapping particles even when they are present in low concentrations, as low as $10^5$ particles per ml. In contrast, without the GET system, we were unable to trap a particle even after waiting for an hour using a regular plasmonic aperture trap for this low particle concentration. Details are provided in Supplementary Discussion VII and Supplementary Movies 10-12.

The GET trap with plasmonic cavities provides multiple advancements over the state-of-the-art plasmonic optical nanotweezers and has many potential ramifications. Of note is that in GET, one does not need to have the ability to image the object being trapped before trapping can occur in contrast to anti-Brownian electrokinetic (ABEL) trap[33]. This is useful for Raman spectroscopy of single trapped

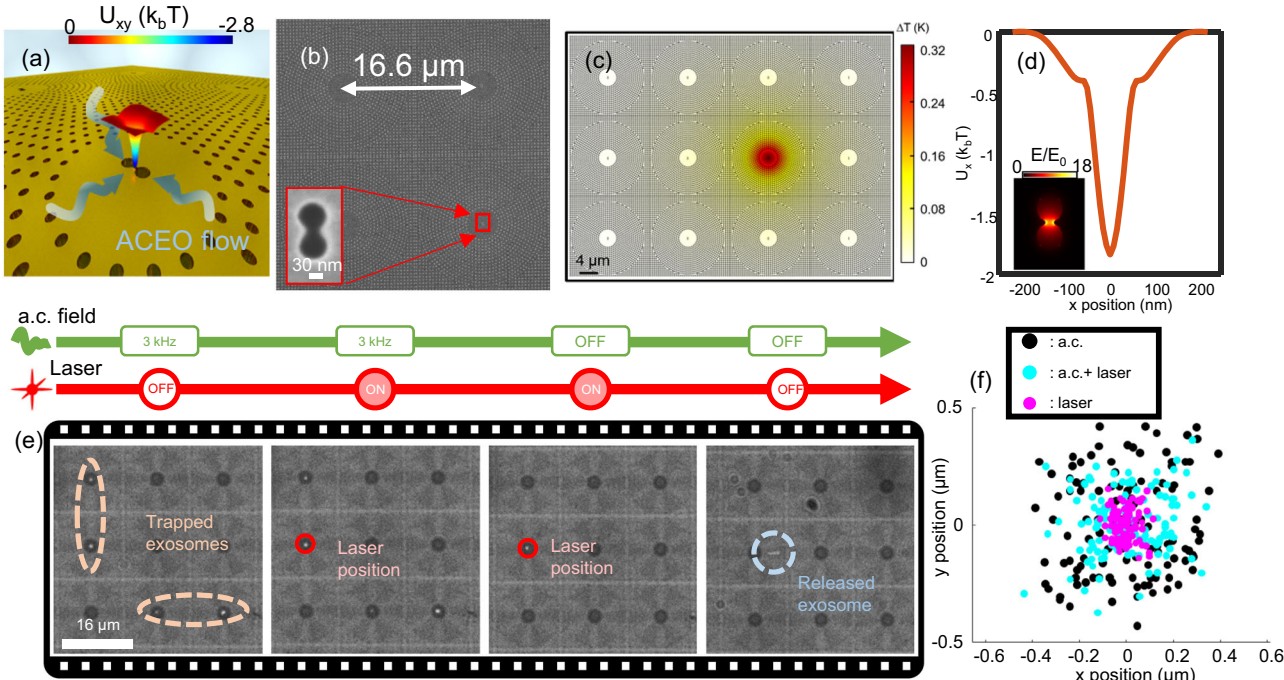

**Fig. 4 | GET with superimposed plasmonic trapping potential and electro-hydrodynamic potential. a** Illustration of the GET system with a plasmonic cavity at the center of the void region. **b** SEM image of the GET trap with a plasmonic double nanohole aperture antenna at the center. The inset illustrates the SEM image of a double nanohole aperture antenna. **c** Temperature field distribution at the surface of the plasmonic double nanohole aperture antenna on the sapphire substrate under the trapping intensity of $3.2 \times 10^9$ W/m² (6.3 mW laser power). The result shows that the temperature rise is negligible when the plasmonic cavity is on the high thermal conductivity sapphire substrate. **d** Simulated optical trapping potential on a 100 nm exosome. The inset shows the electromagnetic field

distribution and enhancement near the double nanohole plasmonic aperture. **e** Frame-by-frame sequence exosome trapping and release using the superposition of electrohydrodynamic and plasmon-enhanced optical trapping potential upon laser illumination (second panel). The laser spot size is 1.6 μm. **f** Scatter plot showing the trapping stability for a single trapped exosome when either the a.c. field is ON (electrohydrodynamic trapping mode), both laser and a.c field are ON, and when only the laser is ON (i.e. plasmonic trapping mode). Results from microfluidic simulation presented in Fig. S2 show the outwards ACEO flow created by the center plasmonic aperture.

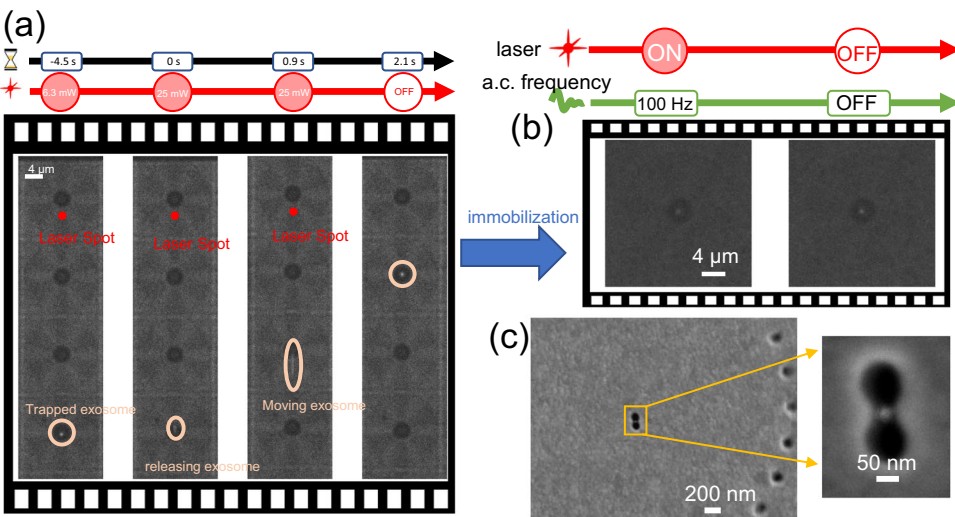

**Fig. 5 | Dynamic manipulation and immobilization of single EV at plasmonic hotspots. a** Frame-by-frame sequence showing the dynamic manipulation of trapped exosomes from one trap to the next for the GET with plasmonic double nanohole aperture depicted in Fig. 4a. Since the plasmonic double nanohole aperture is on a sapphire substrate, a higher laser power of 25 mW is used to induce the ETP flow. The ETP flow perturbs the stagnation zone at nearby GET traps to release a particle from a trap within the region of influence of the ETP flow. Frame-

by-frame demonstration of the transport of trapped single EV from one GET trap to an adjacent trap by using the laser-induced ETP flow. Once the particle has been released and transported near another GET trap by the ETP flow, the laser is then turned OFF so that the a.c. electro-osmotic flow places the particle at the center of the closest GET trap. **b** The trapped single EV is then immobilized by applying a low frequency a.c. field. **c** SEM image of an exosome patterned on the plasmonic double nanohole aperture antenna.

objects because whenever trapping occurs, the particles are always placed at the center of the void region, which is also the location of the plasmonic hotspot automatically. Thus, one can directly perform near-field plasmon enhanced trapping and Raman spectroscopy by illuminating the center of the void region after waiting a few seconds, which is only the time required for the automatic parallelized placement of particles in GET. This capability paves the way for high-throughput plasmon-enhanced trapping and spectroscopies, which has so far remained elusive. Additionally, since each GET trap is spatially different, GET automatically ensures that a different particle can be analyzed at each site, since a different particle is always trapped at each site.

The proposed approach enables the immediate implementation of a myriad of applications including: (1) massive parallel trapping and enhanced spectroscopy of a heterogeneous population of nanosized EVs with single particle selectivity to understand the heterogeneity of EVs, which is a subject of significant scientific interest; (2) ultra-low detection limit nanotweezer diagnostics for liquid biopsy of EVs towards early cancer detection and longitudinal patient treatment monitoring; (3) multifunctional optofluidic molecular selectors for selecting particles of interest postcharacterization; (4) nanotweezer cytometry for profiling size and molecular markers of single trapped EVs. Beyond the field of single EV analysis, this nanotweezer technology may also be utilized for applications in other fields. For example, it can be harnessed for the trapping and analysis of nanoplastics to understand the biochemical properties and the fate of nanoplastics, which are emerging environmental contaminants that are posing environmental concerns. Prior research has reported the use of plasmonic aperture traps for trapping and Raman spectroscopy of engineered nanoparticles[34,35]. The use of plasmonic aperture cavity in a GET system as demonstrated here would enable high throughput trapping even in low particle concentration media and boost the analysis throughput. Finally, with respect to the actively investigated field of quantum photonics, the reported nanotweezer could be harnessed for the parallel placement of multiple quantum emitters and integration with plasmonic cavities to engineer the emission properties of quantum emitters such as generation of high purity entangled photon pairs and the realization of arrays of multiple indistinguishable single photon sources.

## Methods
### Device fabrication
A 10 nm thick Cr mask was deposited on a 1.5 cm × 1.5 cm size silicon wafer using a thermal evaporator. Subsequently, the substrate was spin-coated with PMMA 950 A4 and baked at 180 °C for 2 min. Electron-beam lithography (EBL) was used to pattern the circular nanohole arrays with void regions. The patterned resist was developed in MIBK:IPA 1:3 solution for 35 s, rinsed with Isopropyl Alcohol (IPA), and blown dry with nitrogen. After 4 seconds of descuming, Cr etchant was applied for 10 s to transfer the pattern onto the Cr layer serving as the hard mask for a subsequent reactive ion etching (RIE) process. RIE was proceeded for 1 min to open ~500 nm deep nanoholes into the silicon wafer. To ensure that the whole photoresist and Cr mask layers are stripped off before depositing gold film, the patterned silicon wafer was sonicated in acetone for 10 min, then soaked in Cr etchant for 10 min. Then, the patterned silicon wafer becomes a template. Subsequently, a 120 nm gold film was deposited on the template using an electron beam evaporation machine. We then applied UV-curable epoxy onto the gold film, covered it with an ITO-coated glass substrate, and exposed it under UV light (324 nm wavelength) for 20 min to harden the epoxy. After we peeled the gold film off the Si template, we packed the gold film into a microfluidic channel. The used Si template was cleaned using $O_2$ plasma etching and gold etchant. We reused the Si template by easily depositing another 120 nm thick gold film and performing the template stripping process again. For the experiments with gold nanoholes with centralized plasmonic double nanohole apertures on a sapphire substrate, focused ion beam

milling (FIB) was utilized for the fabrication. The FIB machine was used to drill the desired pattern directly on the gold film on the sapphire substrate.

### Sample preparation
We sandwiched the gold film by covering it with another ITO-coated glass coverslip spaced by a 120 μm thick dielectric spacer to create a microfluidic channel around the patterns. Exosomes were obtained from Creative Diagnostics (DAGA-1003). The exosome solution was fluorescently labelled with FITC dyes that absorb at 488 nm and emit at about 535 nm. The fluorescently labelled polystyrene beads were purchased from Thermo-Fisher Scientific (Fluoro-Max).

### Fluorescence imaging
The trapping and imaging were performed using a custom fluorescent imaging and optical trapping microscope based on a Nikon Ti2-E inverted microscope. The suspended particle solution was injected into the microfluidic channel. A high quantum efficiency sCMOS (Photometrics PRIME 95B) camera was used to acquire images. The plasmonic double nanohole antenna was excited with a 973 nm semiconductor diode laser (Thorlabs CLD1015). The laser beam was focused with a Nikon 40X objective lens (0.75 NA). The a.c. electric field was supplied by a dual-channel function generator (BK Precision 4047B).

### Multiphysics simulations
The electromagnetic simulation was performed using a full-wave simulation formalism in Lumerical FDTD software. Perfectly matched layers were placed at the top and bottom of the domain to prevent backscatter from boundaries. A linearly polarized plane wave served as the light source. A 3D COMSOL model was established to solve the heat transfer and fluid dynamics problem. A prescribed temperature of 293.15 K was set at the boundaries for solving the heat transfer physics. The a.c. electro-osmosis flow was modeled by applying a slip boundary condition on the surface of the nanohole array. The slip velocity is the electro-osmotic slip velocity $\vec{\mathbf{u}}$, which is given by: $\vec{\mathbf{u}} = \mu_{eo}\vec{\mathbf{E}}_t$, where $\mu_{eo} = -\frac{\varepsilon_r \varepsilon_0 \zeta}{\mu}$ is the electro-osmotic mobility, and $\varepsilon_r$ is the relative permittivity. $\varepsilon_0$ is the permittivity of free-space, $\zeta$ is the zeta potential, and $\mu$ is the dynamic viscosity of the liquid. $\vec{\mathbf{E}}_t = \vec{\mathbf{E}} - (\vec{\mathbf{E}} \cdot \vec{\mathbf{n}})\vec{\mathbf{n}}$ and $\vec{\mathbf{E}}$ is calculated by solving the Poisson's equation. The zeta potential used was set to be −33 mV. The no-slip boundary condition, which is $\mathbf{u} = 0$ was set on all other boundaries. The thermal properties of glass, gold, and water were adapted from the COMSOL material library. The relative permittivity of water at a.c. frequencies was set as 78.

## Data availability
The data generated in this study have been deposited in the Harvard Dataverse database under accession code (https://doi.org/10.7910/DVN/JI56ZH).

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

## Acknowledgements

The authors acknowledge financial support from the National Science Foundation NSF CAREER Award (NSF ECCS 2143836) received by J.C.N.

## Author contributions

J.C.N. conceived and guided the project. C.H. fabricated the samples and performed the experiments and the numerical simulations. J.C.N and C.H. discussed the results and wrote the manuscript.

## Competing interests

The Authors declare the following competing interests that C.H. and J.C.N are the inventors on a patent application (No. 63/486,697) submitted by Vanderbilt University. The Authors declare no other competing interests.
