## [Peer Review File · Nature Communications]

Scalable trapping of single nanosized extracellular vesicles using plasmonicsREVIEWER COMMENTS

Reviewer #1 (Remarks to the Author):

In the manuscript: "Scalable trapping of single nanosized extracellular vesicles using plasmonics" the authors present many different results on the trapping, immobilization, transport and isolation of nanoparticles (particularly extracellular vesicles and polystyrene particles) using combinations of electroosmotic, thermal and plasmonic laser tweezers. Their results are very impressive and definitely worthy of consideration for Nature Communications; however, some revisions and perhaps additional measurements are required support the claims of this work and also to explain some of the claims (e.g. is heating really significant for damage? If using fluorescence, is it really any better than a regular optical tweezer? Does dielectrophoresis play a role?).

1) Optical trapping of EVs ~50 nm has been achieved with conventional laser tweezers using fluorescence and/or Raman for identification (<https://doi.org/10.1021/acs.analchem.7b00017>). Is a plasmonic tweezer then required, and if so, what is the advantage? References to this and other existing work on optical tweezers for EVs should be given.

2) It seems that one of the main points of the manuscript is around achieving high throughput but not having substantial plasmonic heating. The authors even endeavor to use high thermal conductivity substrates to reduce heating. At the same time, the temperature increase is only 0.32K and at most 9K; yet, EVs and other biomaterials exist at >10 K above room temperature. Is this really a valid concern then since I am assuming the setup is at room temperature. If this and other works never approach damaging temperatures, then why is this being flagged as an advantage of the work.

3) More details should be presented on the SEM imaging of immobilized particles. How was the sample dried? What was the yield of this process (critical for claims of immobilizing quantum emitters)? Why is the nanoparticle appearing bright (conductor?) in the SEM image – is it not a dielectric?

4) The authors claim generalization of their technique towards nanoplastics, but also for use in Raman or other types of spectroscopies. There is significant literature on spectroscopy of trapped nanoparticles (including Raman of nanoparticles in apertures) that is not discussed. They should include some discussion of the state of the art in this field.

5) Further to the previous point, the authors rely on fluorescence for all their measurements. However, plasmonic traps have used just the scattering changes to monitor trapping events (label-free). This would be critical to support their claim of monitoring nanoplastics or generally EVs that are not labelled. Why do the authors not monitor the trapping laser signal since this has become a standard approach to measuring trapping events for apertures in metal films for other researchers? There is considerable complexity with adding the plasmonics and I do not see the justification for this added complexity if the end result is something that cannot be achieved with conventional tweezers (e.g., label free detection). The authors allude to moving away from fluorescence in the manuscript, but do not support their claim by actually demonstrating it.

6) The ACEO pushes particles away from apertures – does this make it more difficult to trap at the center aperture? It seems this is the case from the distribution in Figure 4c, as the particle is not as localized when the AC is on. What is the laser trapping threshold power with and without the AC? This is critical because having the requirement not to have laser damage is one of the main claims of the manuscript, but requiring a higher power to trap stably goes against this claim.

7) How critical is the usage of ACEO in achieving rapid trapping? For example, on the basis of diffusion alone, how quickly do particles visit the trapping region and can be trapped by the laser? I expect that the time to trap would be around milliseconds to seconds if there is not significant repulsion of the particles due to thermal or electric effects. I think it is critical that the authors show also trapping of the particles with the single central aperture alone as a means to validate their claim that this approach speeds up trapping. A fair comparison would be employing the optimal conditions for each configuration.

8) How do the authors rule out dielectrophoretic effects? I think these are distinct from the ACEO (since charged species is not required) and can also result in repulsion of particles near an aperture. Indeed, DEP has been used by many authors in this area to localize nanoparticles in some geometries, and so a calculation of their relative magnitude should be given to show that ACEO is indeed the dominant force.

9) Some details are missing from the methods – specifically what is the part number of the EVs used, polystyrene used etc.

10) What solution is being used in the experiments? Is it pure water, or a physiological buffer or other? If surfactants are used, they can play a significant role on the thermophoretic properties and should be accounted for (<http://dx.doi.org/10.1021/acs.nanolett.0c03638>).

Reviewer #2 (Remarks to the Author):

In this manuscript, the authors develop a novel geometry-induced electrohydrodynamic tweezers (GET) capable of achieving the fast transport of single nanosized objects, specifically extracellular vesicles (EV) by applying a low-frequency a.c. field and laser illumination at plasmonic hotspots. The system generates multiple electrohydrodynamic potentials allowing for the high-throughput tether-free plasmon-enhanced single EV trapping and spectroscopy, while also mitigating photothermal damage. The final design of the platform consists of plasmonic gold nanoholes arranged in a circular geometry with a central void containing the nanoantenna, a sapphire heat-sink, and ITO electrodes. To my knowledge, the authors are indeed correct that no other approach has achieved the stable nanomanipulation of several particles in parallel within seconds using low average power. The experimental results and supplementary videos are very good and would be of strong interest to the plasmonic and biological communities. However, there are several grammatical errors and unclear sentences that need to be corrected before the article can be considered for publication.

Figure 1a(i) and (ii) captions depict the forces generated when the a.c. is turned on.

However, the authors neither describe the details nor mention the forces generated in-text. Additionally, the authors do not mention the experiments with the laser [Fig. 1a(ii)] till the last subsection, "Scalable plasmonic trapping of EVs with GET." It would be beneficial to reorganize the figures.

The authors provide the dimensions of the gold nanoholes and diameter of void region. However, they should also include the distance between the center of two adjacent void regions/ plasmonic nanoantennas. See Fig. 4 in D. Conteduca et al. "Exploring the Limit of Multiplexed Near-Field Optical Trapping," ACS Photonics 8(7), 2060-2066 (2021).

Main text, line 202: "... 100 Hz was applied to 'print' the trapped EVs in position." The authors should clarify the meaning of "printing."

In main text, line 210: the authors should clarify, "a frame-by-frame sequence of the rapid onset and long-lasting stable parallel trapping an ac field of 3.5 kHz is applied."

The frequency-dependence of the drag force from the a.c. electro-osmotic flow is shown in Fig. 3. The authors should comment on why the dipole-dipole interaction is weakly dependent on frequency.

The supplementary videos and images depict the trapping of nanoscale particles within the circular voids. The authors should provide the accuracy of trapping (percent ratio of particles trapped to that of voids) for 100 nm PS beads and EVs? Additionally, how does this change with different sized particles?

Experimental results display the trapping of EVs for >2min. The authors should comment on how long an EV can be trapped in their system without damage.

In the manuscript the authors trap EVs with sizes ranging from 44 nm -95 nm. The authors should comment on whether these are the maximum and minimum sizes of particles that can be trapped with their system.

The authors should discuss why the laser trapping is more stable compared to the combination of a.c. field + laser.

The authors should comment on why 25 mW was chosen to conduct the dynamic manipulation experiments with GET.

In the supplementary section, the authors should expand on the description of variables in equation (1).

In the supplementary section, α is described as the attenuation coefficient (line 39) and particle polarizability (line 145). Similarly, κ represents the imaginary part of the refractive index of gold (line 41) and the characteristic Debye length (line 155). The authors should be

consistent in labeling variables.

In the main text, the authors state “Early developments of plasmonic tweezers rely on Brownian diffusion to load the trap, which is slow, nondeterministic, and a highly time-consuming process that makes plasmonic tweezers impractical for low particle concentration solutions.” They then jump to discussing ETP a way of using plasmonic heating for trapping. However, there were several prior works that should be cited that leveraged heating for trapping in the absence of an applied a.c. e-field. See for example the following:

*Roxworthy, B. J. et al. Application of plasmonic bowtie nanoantenna arrays for optical trapping, stacking, and sorting. *Nano Lett.* 12, 796–801 (2012).

*Juan, M. L., Righini, M. & Quidant, R. Plasmon nano-optical tweezers. *Nat.*

*Roxworthy, B., Bhuiya, A., Vanka, S. et al. Understanding and controlling plasmon-convection. *Nat Commun* 5, 3173 (2014). <https://doi.org/10.1038/ncomms4173>

Minor comments

In-text mentions of figures should be labelled as “ ... Fig.x” rather than “... Figure x.”

Figure numbers should be bolded within each figure caption

Commas are not appropriate after each subsection within the figure captions. For example, it should be “Figure x. (a) ,(b)... ,” instead of “Figure x. (a), ... (b), ...”

In-text description of figures should follow the order in which the sub-figures are labelled. For eg. the authors should discuss the contents of Fig. 1(c) prior to Fig. 1(d).

Main text, line 35: replace “They compromise exosomes” to “They comprise of exosomes”

Main text, line 122: replace “...excite surface plasmon wave that efficiently...” “...excite surface plasmon waves that efficiently...”

Add axes for Fig. 1 (c).

Main text, line 134: change to “radially outwards”

Figure 1a (i) and (ii) are hardly visible, and could be placed in the supplementary (similar to Fig. S5)

Figure 3: there is no label for part (a) within the figure caption.

Main text, line 321: change to “Fig. 4(a)”

Add scale-bars for Fig. 2(a), Fig. 5(c) inset.

Supplementary, add appropriate punctuations (periods or commas) after equations. For eg, there should be a period after equation (1), comma after equation (4), etc.

Supplementary, line 59: change to: “the ETP flow arises”

We thank the reviewers for their meticulous review of our manuscript and for providing multiple constructive comments to improve the quality of our manuscript. We have carefully addressed the reviewers' comments and have performed additional experiments where necessary. We have provided below a detailed response to the reviewers' comments.

REVIEWER COMMENTS

Reviewer #1 (Remarks to the Author):

In the manuscript: "Scalable trapping of single nanosized extracellular vesicles using plasmonics" the authors present many different results on the trapping, immobilization, transport and isolation of nanoparticles (particularly extracellular vesicles and polystyrene particles) using combinations of electroosmotic, thermal and plasmonic laser tweezers. Their results are very impressive and definitely worthy of consideration for Nature Communications; however, some revisions and perhaps additional measurements are required support the claims of this work and also to explain some of the claims (e.g. is heating really significant for damage? If using fluorescence, is it really any better than a regular optical tweezer? Does dielectrophoresis play a role?).

We thank the reviewer for the positive comments on our manuscript. We have carefully addressed the reviewer's comments in detail below.

1) Optical trapping of EVs ~50 nm has been achieved with conventional laser tweezers using fluorescence and/or Raman for identification (<https://doi.org/10.1021/acs.analchem.7b00017>). Is a plasmonic tweezer then required, and if so, what is the advantage? References to this and other existing work on optical tweezers for EVs should be given.

Response: We thank the reviewer for this question. Indeed, there are a few papers that have reported the use of conventional optical tweezers to trap EVs and perform Raman spectroscopy, including the paper mentioned by the reviewer. There are several limitations of the optical tweezer approach that make having plasmonic tweezers necessary. First, the optical tweezer approaches require laser powers of at least 100 mW to trap the EVs¹⁻⁴. Please see refs 1-4 below. In some of these studies, the researchers have reported that this high laser power requirement often induces explosion and damage to trapped EVs. For example, Kruglik et al.¹ reported using 100 mW laser power for trapping extracellular vesicles and performing Raman spectroscopy and they reported laser-induced damage in the optical trap. Here is a quote from their paper: ***"The main damage due to the high power density of light in the optical trap might consist of sudden explosions of the trapped vesicles; the debris are then moved immediately away from the trap via Brownian motion."***¹.

The only way to reduce the potential for damage in optical tweezers is to reduce the laser power used for trapping, but this comes at the cost of reducing the trapping stability and reducing the signal-to-noise ratio for Raman acquisition. For example, the authors attempted to lower the laser power to 25 mW but they experienced dramatically reduced trapping stability. In fact, they made the following statements in their paper with respect to using a low power of 25 mW. ***"An additional issue we encountered was the tendency of trapped particles to fall out of the trap before completion."***⁴.

Another disadvantage of laser-tweezer based Raman is that the laser beam typically traps multiple EVs within the diffraction limited focused laser spot to be able to obtain Raman signals for the nanoscale

EVs. This is because Raman signals are very weak, and in an optical trap, one needs to capture many particles to obtain sufficient signals.

Plasmonic nanotweezers, on the other hand, generate strong electric field gradients and enhance the local field intensity. Since the gradient trapping force is proportional to intensity gradients, these features of plasmonic tweezers ensure they can provide stable trapping and the enhancement of the spontaneous Raman signals from single trapped nanoscale EVs. Plasmonic tweezers also ensure single EV trapping since the plasmonic hotspot is nanolocalized to support the trapping of a single EV.

The following statements have been included from line 49 to line 57 of the manuscript to clarify these points:

“Optical tweezer based Raman spectroscopy requires high laser power (~ 100 mW or higher) to stably trap EVs¹⁻⁴, which sometimes leads to the explosion of EVs that interferes with subsequent measurements¹. Lowering the laser power brings about multiple issues including insufficient trapping stability so that the particle escapes from the trap before the completion of the signal collection and restriction on the size of vesicles that can be trapped⁴. Furthermore, optical tweezers do not guarantee single EV trapping and often result in the collection of multiple EVs within the diffraction-limited laser spot. Finally, the loading of EVs into the optical trap is a slow process that can take several minutes, which will adversely impact the analysis throughput¹.”

2) It seems that one of the main points of the manuscript is around achieving high throughput but not having substantial plasmonic heating. The authors even endeavor to use high thermal conductivity substrates to reduce heating. At the same time, the temperature increase is only 0.32K and at most 9K; yet, EVs and other biomaterials exist at >10 K above room temperature. Is this really a valid concern then since I am assuming the setup is at room temperature. If this and other works never approach damaging temperatures, then why is this being flagged as an advantage of the work.

Response: We thank the reviewer for this comment! We fully agree with the reviewer that EVs and other biomaterials exist at >10 K above room temperature since the human body temperature is around 37 °C, which is 10 to 15 °C above room temperature. Hence, the range of temperatures achieved in our work is all safe for EVs.

The primary reason for using sapphire as a substrate to achieve a negligible temperature rise of only 0.32 K instead of 2.7 K (achievable with glass substrate, as depicted in Fig. S4) is to preclude any ETP flow when we are illuminating a specific plasmonic aperture at a given GET site. This would enable one to probe a given trap without any induced ETP flow that may influence the particle dynamics in a nearby GET trap in the array. By suppressing the temperature rise to 0.32 K using a sapphire substrate, we ensure that ETP flow is negligible and does not perturb the particles trapped at nearby GET sites. Thus, particles at nearby traps would not be released. To regain ETP flow for dynamic manipulation on demand, the laser power can be increased well-above the threshold needed for stable nanoscale optical trapping and made to illuminate the surrounding nanohole array to induce ETP flow as needed. Another advantage of using the sapphire substrate to dissipate excess heat is to ensure that plasmonic heating does not increase the Brownian diffusion of the particles, which would impact the trapping stability during the plasmon-assisted trapping process.

We have revised the main manuscript and made the following statements from line 254 in the manuscript to clarify this point:

“Fig. 4c shows that the local temperature rise for an incident intensity of $3.2 \times 10^9 \text{ W/m}^2$ (laser power of 6.3 mW) is only 0.32 K. This negligible temperature rise not only prevents trapped exosome from heating-induced damage, but also primarily suppresses the generation of thermal-related effects that destabilize the trapping, such as positive thermophoresis or convection. Furthermore, the low temperature rise achievable with a sapphire substrate suppresses ETP flow from destabilizing the particles trapped at nearby GET sites when another GET site is illuminated.”

3) More details should be presented on the SEM imaging of immobilized particles. How was the sample dried? What was the yield of this process (critical for claims of immobilizing quantum emitters)? Why is the nanoparticle appearing bright (conductor?) in the SEM image – is it not a dielectric?

Response: To dry the sample, we first opened the microfluidic channel by removing the ITO-coated cover slip. Then we gently removed the residual fluid by blowing it with nitrogen. Subsequently, the chip was imaged with SEM.

For the yield of the process, we need to clarify here that to immobilize the particle precisely into the hotspot, we need both laser illumination and ultra-low frequency a.c. field. After the particle had been trapped by the plasmonic aperture, we then introduced the ultra-low frequency a.c. electric field to place the particle into the cavity, while the laser was still on. By having the laser on prior to applying the low frequency a.c. field, it ensures that the particle is precisely held at the plasmonic hotspot prior to the immobilization. This sequence ensures that immobilization always happened at the plasmonic aperture.

The nanoparticle is dielectric because it is a vesicle with a lipid-bilayer. We used the secondary-electron detector of the SEM, and we believe the dielectric particle appears bright because it is an insulator and charges are accumulating on its surface. We refer to the following references where EVs were observed under an SEM and the EVs were shown to appear bright in concurrence with our observations^{5,6}.

4) The authors claim generalization of their technique towards nanoplastics, but also for use in Raman or other types of spectroscopies. There is significant literature on spectroscopy of trapped nanoparticles (including Raman of nanoparticles in apertures) that is not discussed. They should include some discussion of the state of the art in this field.

Response: We thank the reviewer for these comments. We have added the following statements in the main manuscript to discuss prior work on Raman characterization of nanoparticles.

The following discussion regarding the current state-of-art in the field of spectroscopy with trapping has been added to the main manuscript:

Line 49: “Optical tweezer based Raman spectroscopy requires high laser power (~ 100 mW or higher) to stably trap EVs¹⁻⁴, which sometimes leads to the explosion of EVs that interferes with subsequent measurements¹. Lowering the laser power brings about multiple issues including insufficient trapping stability so that the particle escapes from the trap before the completion of the signal collection and restriction on the size of vesicles that can be trapped⁴. Furthermore, optical tweezers do not guarantee

single EV trapping and often result in the collection of multiple EVs within the diffraction-limited laser spot. Finally, the loading of EVs into the optical trap is a slow process that can take several minutes, which will adversely impact the analysis throughput¹.”

Line 362: “Prior research have reported the use of plasmonic aperture traps for trapping and Raman spectroscopy of engineered nanoparticles^{7,8}. The use of plasmonic aperture cavity in a GET system as demonstrated here would enable high throughput trapping even in low particle concentration media and boost the analysis throughput.”

5) Further to the previous point, the authors rely on fluorescence for all their measurements. However, plasmonic traps have used just the scattering changes to monitor trapping events (label-free). This would be critical to support their claim of monitoring nanoplastics or generally EVs that are not labelled. Why do the authors not monitor the trapping laser signal since this has become a standard approach to measuring trapping events for apertures in metal films for other researchers? There is considerable complexity with adding the plasmonics and I do not see the justification for this added complexity if the end result is something that cannot be achieved with conventional tweezers (e.g., label free detection). The authors allude to moving away from fluorescence in the manuscript, but do not support their claim by actually demonstrating it.

Response: We thank the reviewer for this comment. We fully agree that the label-free technique has been used for monitoring trapping events^{9,10}. But we also note that fluorescence imaging is another approach for monitoring the trapping particle, and it is very straightforward, and enables us to visualize the particles in real-time and analyze their behavior during trapping. The following references have reported the use of fluorescence imaging to characterize trapping including in aperture tweezers^{11–16}. Our GET system is also designed to enhance the fluorescence signal from the trapped particles, which is discussed in detail in the SI (section III). By transforming the nanohole array lattice into a radial lattice, we also showed GET’s ability to efficiently beam up the emitted fluorescence light into a small divergence angle, for enhancing the signal collection.

Our reference to using GET for label-free spectroscopy relates to Raman spectroscopy. The GET system with a plasmonic cavity can be used to enhance Raman signals after high throughput particle trapping. For Raman enhancement, the particle needs to be placed at the region of highest field enhancement. We already showed that we can precisely place a particle within the nanogap of a plasmonic aperture cavity, which is where maximum Raman enhancement can be expected as shown in Fig. 5c of the main manuscript.

In summary, the key highlight of GET with central plasmonic cavities is that it provides a new platform that enables to trap single particles in parallel near the vicinity of a plasmonic cavity so that the illumination of any plasmonic cavity then stably traps the particle instantaneously precisely at the plasmonic cavity using plasmon-enhanced force. There is a no need to wait for slow Brownian diffusion to load the trap because in the GET system, particles are already placed near the plasmonic cavities in parallel within seconds. This is extremely important especially when working in diluted particle concentration media as depicted in Fig. S11.

6) The ACEO pushes particles away from apertures – does this make it more difficult to trap at the

center aperture? It seems this is the case from the distribution in Figure 4c, as the particle is not as localized when the AC is on. What is the laser trapping threshold power with and without the AC? This is critical because having the requirement not to have laser damage is one of the main claims of the manuscript, but requiring a higher power to trap stably goes against this claim.

Response: The ACEO from the center aperture does have its own effect on the optical trap, as we illustrated in Fig. 4f of the main manuscript. The trapping stability is maximized when only the laser was turned on (after transporting the particle), and we attribute the expansion of the particle position distribution to the existence of ACEO flow from the center aperture. But the existence of ACEO does not necessarily make the trapping more difficult from happening, because the ACEO flows from the nanohole array are so strong that they push the particle to the very vicinity of the plasmonic cavity. After the particle had been brought close to the center aperture cavity, the laser can then be introduced and we observed the trapping stability significantly increased, indicating the occurrence of near-field trap. So, even with no laser illumination, the particle will not escape and is always ready for near-field optical trapping. We believe this is the most significant highlight of our GET system, namely that GET ensures that particles are “waiting” in parallel near each of the center aperture cavity to be plasmonically trapped.

Regarding the trapping threshold without the AC field, the trapping threshold is largely dependent on the size of trapped EV or particle. We ran full-wave simulation to calculate the optical trapping potential, similar to the one illustrated in Fig. 4d. In Fig. 4d, we obtained 1.8 k_BT trapping potential with 3.2×10⁹ W/m² (6.3 mW laser power). For a 100 nm exosome, the calculated trapping potential along the x direction drops to 1 k_BT after the laser intensity reduces to 1.78×10⁹ W/m² (3.5 mW laser power), where we believe is close to the lower limit of trapping laser intensity.

7) How critical is the usage of ACEO in achieving rapid trapping? For example, on the basis of diffusion alone, how quickly do particles visit the trapping region and can be trapped by the laser? I expect that the time to trap would be around milliseconds to seconds if there is not significant repulsion of the particles due to thermal or electric effects. I think it is critical that the authors show also trapping of the particles with the single central aperture alone as a means to validate their claim that this approach speeds up trapping. A fair comparison would be employing the optimal conditions for each configuration.

Response: We thank the reviewer for this crucial question. Relying on diffusion alone, the waiting time before the trapping event happens can be uncertain and long. For example, Jones et al.¹⁷ stated in prior literature: ***“Typically, the initial trapping event will occur anywhere from several minutes to a few hours after the system has been setup and stabilized”***.

We also theoretically calculated the time for free diffusion of 100 nm particles in water suspension under various particle concentrations. The environmental temperature was assumed to be 300 K. The average distance separating nano-particles in a water suspension with concentration of C particles/volume (for SI unit) can be expressed by¹⁸:

$$L = 1/\sqrt[3]{C}$$

Relying on free diffusion, we estimate the time for a 100 nm particle to cover this distance. The mean squared diffusion distance is given by:

$$L^2 = 2D\tau ;$$

τ is the estimated time and D is the diffusion coefficient, which is given by:

$$D = \frac{K_B T}{6\pi\eta r}$$

K_B stands for the Boltzmann's constant, T is the environmental temperature, η is the viscosity of water at the specific temperature and r is the radius of nanoparticle.

As depicted in the following, under the low concentration used in some of our experiments, for example, 1×10^5 particles/mL, the estimated loading time is as long as 9404 s (156.74 minutes). But with ACEO flow in a GET platform, under this low concentration, we can still observe particle trapping by the electrohydrodynamic potentials within seconds, which makes it ready to be captured by a plasmonic aperture under laser illumination.

We appreciate the suggestion from the reviewer about additional experimental data to 'show also trapping of the particles with the single central aperture alone'. We have performed additional experiments to compare trapping with and without GET speed up for varying particle concentrations and this information is provided in the SI.

As shown in Fig. S11 and SI video7, using the solution of 100 nm polystyrene beads with concentrations of 10^5 , 10^6 , 10^8 and 10^9 particles/mL, we found that GET can initiate trapping under all concentrations within seconds. When the concentration of particle is as low as 10^5 particles/ml, the estimated time is 157 minutes by assuming a free diffusion under Brownian motion. In the experiment, we used the laser spot to illuminate one of the plasmonic cavities for one hour (without any ACEO flow) and did not observe any trapping. As a comparison, we also conducted experiment using GET to initiate trapping in solutions for varying concentrations of 10^5 , 10^6 , 10^8 and 10^9 particles/ml. GET rapidly loads particles into the traps within 5 seconds under all the concentrations we examined.

Fig. S11: the average diffusion time obtained from theoretical estimation (blue), single plasmonic cavity (red) and GET system (black). It clearly shows the GET can rapidly transport particles towards trapping site even at 0.16 femtomolar (10^5 particles/mL) concentration, which is not feasible for traditional optical tweezers or near-field tweezers.

This set of experimental data validates the superiority of our GET system on speeding up the loading process for low particle concentration solution.

The following statements have also been added on line 297 in the main manuscript:

“We conducted further experiments to compare the effectiveness of plasmonic trapping with and without the GET system. Our findings indicate that the GET system is capable of trapping particles even when they are present in low concentrations, as low as 10^5 particles per ml. In contrast, without the GET system, we were unable to trap a particle even after waiting for an hour using a regular plasmonic aperture trap for this low particle concentration. Details are provided in section VII of the supplementary information.”

8) How do the authors rule out dielectrophoretic effects? I think these are distinct from the ACEO (since charged species is not required) and can also result in repulsion of particles near an aperture. Indeed, DEP has been used by many authors in this area to localize nanoparticles in some geometries, and so a calculation of their relative magnitude should be given to show that ACEO is indeed the dominant force.

Response: We thank the reviewer for this interesting question. We have carried out numerical simulations in COMSOL Multiphysics to show the negligible contribution of DEP effects. This information has been added to the SI section VII. The DEP force is proportional to the volume of the particle and hence becomes very weak for very small particles. DEP may also be either positive DEP (particles are attracted to a high field region) or negative DEP (particles are repelled from the high field region) depending on the sign of the Clausius-Mossotti factor.

The DEP potential across the center of the void region is shallow compared with optical potential or EHD potential, and it is only $0.03 k_bT$ on 100 nm polystyrene bead. Meanwhile, the EHD potential under the same condition shows a $15 k_bT$ depth. We thus conclude that ACEO flow is indeed dominating.

Figure S12: (a) potential from DEP. (b) SEM of one GET trap. The blue line and red line show the positions where we extract the trapping potential in (a) and (c), respectively. (c) is the trapping potential from ACEO drag force.

Figure S13: the colormap showing the in-plane DEP force experienced by a 100 nm polystyrene bead positioned 50 nm above the gold surface, calculated using equation 15.

To clarify this, we added Fig. S12 and S13 as well as the statements in the SI section VIII:

“Dielectrophoresis (DEP) is the phenomenon where a dielectric particle experiences a force in a non-uniform electric field. Here, as the nanohole array perturbs the a.c. electric field to create ACEO flow, the distorted a.c. electric field may also exert DEP force onto trapped particles. DEP force is governed by:

$$F_{DEP} = 2\pi a^3 \epsilon_m \text{Re} \left(\frac{\epsilon_p^* - \epsilon_m^*}{\epsilon_p^* + 2\epsilon_m^*} \right) \nabla |E_{rms}|^2, \quad (15)$$

where $\epsilon_p^* = \epsilon_p + \frac{S_p}{i\omega}$ and $\epsilon_m^* = \epsilon_m + \frac{S_m}{i\omega}$. ϵ_m and ϵ_p are the values of permittivity of the medium and particle, respectively. S_m and S_p are the conductivities of medium and particles, respectively. E_{rms} is the root-mean-square value of the applied a.c. electric field and ω is the a.c. frequency. The term $\frac{\epsilon_p^* - \epsilon_m^*}{\epsilon_p^* + 2\epsilon_m^*}$ is also defined as Clausius-Mossotti factor (CM factor), which decides if the DEP is repulsive or attractive. The value of CM factor versus a.c. frequency is obtained from prior research¹⁹, resulting in a shallow potential of 0.03 $k_B T$ from DEP. Compared with the ACEO potential from Stoke’s drag, the influence from DEP can be ruled out from this analysis.

Since the plasmonic cavity (DNH) may slightly perturb the a.c. electric field, we also studied the DEP force created in its presence. Using the same method described in SI section I for calculating the a.c. electric field, we map the in-plane DEP force exerted on a 100 nm polystyrene bead positioned 50 nm above the gold film using equation (15). The colormap in Fig. S13 reveals that the DEP force is ~ 1.5 fN, which is negligible.”

9) Some details are missing from the methods – specifically what is the part number of the EVs used, polystyrene used etc.

Response: We have provided information about the EV and polystyrene part number in the Method section. The EVs are FITC-conjugated exosomes that are purchased from Creative Diagnostics (DAGA-1003), while the polystyrene beads are purchased from Thermofisher Scientific (Fluoro-Max).

10) What solution is being used in the experiments? Is it pure water, or a physiological buffer or other? If surfactants are used, they can play a significant role on the thermophoretic properties and should be accounted for (<http://dx.doi.org/10.1021/acs.nanolett.0c03638>).

Response: We thank the reviewer for this question. In the experimental demonstration, the solution was DI water, and no surfactant has been used.

Reviewer #2 (Remarks to the Author):

In this manuscript, the authors develop a novel geometry-induced electrohydrodynamic tweezers (GET) capable of achieving the fast transport of single nanosized objects, specifically extracellular vesicles (EV) by applying a low-frequency a.c. field and laser illumination at plasmonic hotspots. The system generates multiple electrohydrodynamic potentials allowing for the high-throughput tether-free plasmon-enhanced single EV trapping and spectroscopy, while also mitigating photothermal damage. The final design of the platform consists of plasmonic gold nanoholes arranged in a circular geometry with a central void containing the nanoantenna, a sapphire heat-sink, and ITO electrodes. To my knowledge, the authors are indeed correct that no other approach has achieved the stable nanomanipulation of several particles in parallel within seconds using low average power. The experimental results and supplementary videos are very good and would be of strong interest to the plasmonic and biological communities. However, there are several grammatical errors and unclear sentences that need to be corrected before the article can be considered for publication.

1) Figure 1a(i) and (ii) captions depict the forces generated when the a.c. is turned on. However, the authors neither describe the details nor mention the forces generated in-text. Additionally, the authors do not mention the experiments with the laser Fig. 1a(ii)] till the last subsection, “Scalable plasmonic trapping of EVs with GET.” It would be beneficial to reorganize the figures.

Response: We thank the reviewer for this comment. We have redrawn the image to not have the double nanohole aperture, as shown below and in Fig. 1a of the main context. The GET system with double nanohole now appears only in Fig. 4 for ease of flow.

Figure 1: Illustration and theoretical analysis of the GET system. (a) Illustration of the operating mechanism of the GET system. The tangential a.c. field induces electro-osmotic flow that is radially outward. By harnessing a circular geometry with a void region, the radially outward a.c. electro-osmotic flow creates a stagnation zone at the center of the void region where trapping takes place. (b) The evolution from a square-lattice nanohole array into a radial-lattice nanohole array. (c) Radiation energy flow for a dipole fluorescence emitter placed at the center of the void region showing the ability to harness the GET trap to also beam emitted photons from trapped particles. (d) COMSOL simulation of the radial electro-osmotic flow showing that the geometry of the void region results in opposing electro-osmotic flow that forms a stagnation zone at the center. Particle trapping occurs at the center of the void region where the flow vectors converge. Particle trapping position is highlighted with green dots, (e) SEM image of the plasmonic metasurface array with void regions. Each void region represents a GET trap and can be readily scaled from hundreds to thousands or millions as desired.

2) The authors provide the dimensions of the gold nanoholes and diameter of void region. However, they should also include the distance between the center of two adjacent void regions/ plasmonic nanoantennas. See Fig. 4 in D. Conteduca et al. "Exploring the Limit of Multiplexed Near-Field Optical Trapping," ACS Photonics 8(7), 2060-2066 (2021).

Response: The center-to-center distance of two adjacent voids was $16.67\ \mu\text{m}$. We have also added the SEM images showing the distance between the plasmonic cavities in Fig. 4b.

Figure 4. (b) SEM image of the GET trap with a plasmonic double nanohole aperture antenna at the center.

3) Main text, line 202: "... 100 Hz was applied to 'print' the trapped EVs in position." The authors should clarify the meaning of "printing."

Response: We thank the review for this comment. 'Print' was used to mean immobilizing the particle on to the substrate. We have rephrased this sentence on line 207 of the revised manuscript as: "100 Hz was applied to immobilize the trapped EVs in position".

4) In main text, line 210: the authors should clarify, "a frame-by-frame sequence of the rapid onset and long-lasting stable parallel trapping an ac field of 3.5 kHz is applied."

Response: The statement is rephrased into "A frame-by-frame sequence of the rapid-onset (within 3 seconds) and long-lasting (>2 min) stable parallel trapping when an a.c. field of 3.5 kHz is applied."

5) The frequency-dependence of the drag force from the a.c. electro-osmotic flow is shown in Fig. 3. The authors should comment on why the dipole-dipole interaction is weakly dependent on frequency.

Response: We attribute this to the fact that when the a.c. frequency is low, the surface charge and double layer charge around the particle have sufficient time to respond to and be polarized by the field.

The following statement has been added on SI section V, line 152, to emphasize this point: "We attribute this weak dependence to the fact that when the a.c. frequency is low, the surface charge and double layer charge of the trapped particle have sufficient time to respond to and be polarized by the external a.c. electric field."

6) The supplementary videos and images depict the trapping of nanoscale particles within the circular voids. The authors should provide the accuracy of trapping (percent ratio of particles trapped to that of voids) for 100 nm PS beads and EVs? Additionally, how does this change with different sized particles?

Response: The percent ratio of particles trapping, as being called 'filling factor' is highly dependent on

the concentration of particle. Additional experiments have been conducted to illustrate the relationship between filling factor and the concentration of the particles.

The following statements have also been added on line 176 in the main manuscript:

“This rapid loading and trapping of nanosized particles in parallel is achieved for all the particle concentrations we examined (10^5 to 10^9 particles/ml). Details of the results are discussed in supplementary section IV.”

Results are added to the SI section IV. As shown below, as the particle concentration increases, the number of occupied traps increases.

Figure S7: (a) to (d), the fluorescence images showing the number of occupied GET traps under various 100 nm PS beads concentrations of 10^5 , 10^6 , 10^8 and 10^9 particles/mL, correspondingly. As the concentration of particle increases, more traps are occupied.

With respect to the size of particles, we do not observe the ‘filling factor’ varies with using different sizes of particles. We observed that most of the traps are occupied with the same concentration of 20 nm and 100 nm polystyrene beads (10^8 particles/ml), as shown in Fig. S7(c) and Fig. S10.

7) Experimental results display the trapping of EVs for >2min. The authors should comment on how long an EV can be trapped in their system without damage.

Response: We conducted the experiment showing the trapped particle kept undamaged (fluorescence remained unchanged) after over 25 mins of trapping. The video is shown in supplementary video 13. We expect that the EVs can be trapped indefinitely for as long as needed since there is no thermal heating from the applied a.c. field.

This frame sequence demonstrates the rapid load to a plasmonic cavity and near-field trapping on the plasmonic cavity. After the particle was captured, we kept the laser on and observed the particle after 25 min. The fluorescence emission did not change after 25 min trapping, indicating no structural damage on the particle induced by the near-field trapping.

8) In the manuscript the authors trap EVs with sizes ranging from 44 nm -95 nm. The authors should comment on whether these are the maximum and minimum sizes of particles that can be trapped with their system.

Response: The EV sizes ranging from 44 nm to 95 nm that were trapped is not the minimum and maximum size that can be trapped. This happened to be the range of EV sizes in our sample solution since nanoscale EVs are heterogeneous and range from 30 nm to 130 nm.

We have also performed experiments where we trapped polystyrene beads of 20 nm and 200 nm diameter with GET. Frames sequences showing trapping of 200 nm and 20 nm beads are provided in Fig. S9 and S10, respectively. The videos are included in Supplementary video 8 and 9. With these videos, they should suffice to support that GET can trap particles with size ranging from 30 nm to 130 nm corresponding to EVs.

We added following content in the supplementary information section VI to clarify:

“Initially, we test on PS beads with 200 nm in diameter, which is beyond the upper bound of commonly-known size of EVs. Fig. S9 clearly shows the capability of GET to capture 200 nm polystyrene beads.

Figure S9: 200 nm polystyrene beads are trapped by GET under 3 kHz a.c. frequency.

As illustrated in Fig. S10, after 3 kHz a.c. electric field was turned on, 20 nm PS beads are trapped by the array of electrohydrodynamic potentials. The concentration of 20 nm PS beads is 10^8 particles/mL in the demonstration. Thus, most of the cavities are occupied by the PS beads.

Figure S10: 20 nm polystyrene beads are trapped by GET under 3 kHz a.c. frequency.”

9) The authors should discuss why the laser trapping is more stable compared to the combination of a.c. field + laser.

Response: When we have a.c. and laser both on, the local ACEO flow from the double nanohole aperture may impact additional drag force on the particle directed away from the aperture and reduce the trapping stability. But the ACEO flows from the nanohole array is so strong that it still pushes particles inwards into the very vicinity of the plasmonic cavity. The laser then was introduced to optically trap particles.

To elaborate on the ACEO from the single double nanohole, we conducted numerical simulation in COMSOL to show the ACEO flow profile near a double nanohole aperture within the GET site. The simulation result is added to SI Fig. S2.

Figure S2: in plane ACEO flow profile 10 nm above gold film with DNH, showing the DNH has its own local ACEO flow to push outwards. Thus, the trapping stability is maximized when only laser was on as shown in Fig. 4f.

10) The authors should comment on why 25 mW was chosen to conduct the dynamic manipulation experiments with GET.

Response: It is because when we use high thermal conductivity substrate, sapphire, we need relatively high laser power to generate enough temperature rise for the ETP flow. 25 mW was the lowest power we observed to successfully remove the particle from the ACEO electrohydrodynamic potential.

11) In the supplementary section, the authors should expand on the description of variables in equation (1).

Response: The description is added in SI section I as: “ ρ is the total volume charge density and ϵ is the permittivity of the medium. \mathbf{E} is the electric field and V is the electric potential energy.”.

12) In the supplementary section, α is described as the attenuation coefficient (line 39) and particle polarizability (line 145). Similarly, κ represents the imaginary part of the refractive index of gold (line 41) and the characteristic Debye length (line 155). The authors should be consistent in labeling variables.

Response: Modification has been applied. Thanks for this suggestion.

13) In the main text, the authors state “Early developments of plasmonic tweezers rely on Brownian diffusion to load the trap, which is slow, nondeterministic, and a highly time-consuming process that makes plasmonic tweezers impractical for low particle concentration solutions.” They then jump to discussing ETP a way of using plasmonic heating for trapping. However, there were several prior works that should be cited that leveraged heating for trapping in the absence of an applied a.c. e-field. See for example the following:

*Roxworthy, B. J. et al. Application of plasmonic bowtie nanoantenna arrays for optical trapping, stacking, and sorting. *Nano Lett.* 12, 796–801 (2012).

*Juan, M. L., Righini, M. & Quidant, R. Plasmon nano-optical tweezers. *Nat.*

*Roxworthy, B., Bhuiya, A., Vanka, S. et al. Understanding and controlling plasmon-convection. *Nat Commun* 5, 3173 (2014). <https://doi.org/10.1038/ncomms4173>

Response: Thanks for the note. We have added discussion on other heat-mediated loading methods, at line 64: “Prior strategies using convection flows and thermophoresis to load the trap suffer from particle aggregation and are not suitable for single particle trapping”.

Minor comments

In-text mentions of figures should be labelled as “... Fig.x” rather than “... Figure x.”

Modified.

Figure numbers should be bolded within each figure caption

Modified.

Commas are not appropriate after each subsection within the figure captions. For example, it should be “Figure x. (a) ,(b)... ,” instead of “Figure x. (a), ... (b), ...”

Modified.

In-text description of figures should follow the order in which the sub-figures are labelled. For eg. the authors should discuss the contents of Fig. 1(c) prior to Fig. 1(d).

Modified.

Main text, line 35: replace “They compromise exosomes” to “They comprise of exosomes”

Modified.

Main text, line 122: replace “...excite surface plasmon wave that efficiently...” “...excite surface plasmon waves that efficiently...”

Modified.

Add axes for Fig. 1 (c).

Axes added.

Main text, line 134: change to “radially outwards”

Modified.

Figure 1a (i) and (ii) are hardly visible, and could be placed in the supplementary (similar to Fig. S5)

We have moved Fig. 1a(i) and (ii) into the SI.

Figure 3: there is no label for part (a) within the figure caption.

Added.

Main text, line 321: change to “Fig. 4(a)”

Changed.

Add scale-bars for Fig. 2(a), Fig. 5(c) inset.

Added.

Supplementary, add appropriate punctuations (periods or commas) after equations. For eg, there should be a period after equation (1), comma after equation (4), etc.

Modified.

Supplementary, line 59: change to: “the ETP flow arises”

Changed.

1. Kruglik, S. G. *et al.* Raman tweezers microspectroscopy of *circa* 100 nm extracellular vesicles. *Nanoscale* **11**, 1661–1679 (2019).
2. Lee, W. *et al.* Label-Free Prostate Cancer Detection by Characterization of Extracellular Vesicles Using Raman Spectroscopy. *Anal. Chem* **90**, 48 (2018).
3. Penders, J. *et al.* Single Particle Automated Raman Trapping Analysis of Breast Cancer Cell-Derived Extracellular Vesicles as Cancer Biomarkers. *ACS Nano* **15**, 18192–18205 (2021).
4. Carney, R. P. *et al.* Multispectral Optical Tweezers for Biochemical Fingerprinting of CD9-Positive Exosome Subpopulations. *Anal Chem* **89**, 5357–5363 (2017).

5. Im, H. *et al.* Label-free detection and molecular profiling of exosomes with a nano-plasmonic sensor. *Nat Biotechnol* **32**, 490–495 (2014).
6. Wunsch, B. H. *et al.* Nanoscale lateral displacement arrays for the separation of exosomes and colloids down to 20 nm. *Nat Nanotechnol* **11**, 936–940 (2016).
7. Burkhartsmeyer, J., Wang, Y., Wong, K. S. & Gordon, R. Optical Trapping, Sizing, and Probing Acoustic Modes of a Small Virus. *Applied Sciences* **10**, 394 (2020).
8. Wheaton, S., Gelfand, R. M. & Gordon, R. Probing the Raman-active acoustic vibrations of nanoparticles with extraordinary spectral resolution. *Nat Photonics* **9**, 68–72 (2015).
9. Kotnala, A. & Gordon, R. Quantification of High-Efficiency Trapping of Nanoparticles in a Double Nanohole Optical Tweezer. *Nano Lett* **14**, 853–856 (2014).
10. Pang, Y. & Gordon, R. Optical trapping of a single protein. *Nano Lett* (2012) doi:10.1021/nl203719v.
11. Ndukaife, J. C. *et al.* Long-range and rapid transport of individual nano-objects by a hybrid electrothermoplasmonic nanotweezer. *Nat Nanotechnol* **11**, 53–59 (2016).
12. Wang, K., Schonbrun, E., Steinvurzel, P. & Crozier, K. B. Trapping and rotating nanoparticles using a plasmonic nano-tweezer with an integrated heat sink. *Nat Commun* **2**, 469 (2011).
13. Xu, Z., Song, W. & Crozier, K. B. Direct Particle Tracking Observation and Brownian Dynamics Simulations of a Single Nanoparticle Optically Trapped by a Plasmonic Nanoaperture. *ACS Photonics* **5**, 2850–2859 (2018).
14. Tsuboi, Y. *et al.* Optical trapping of quantum dots based on gap-mode-excitation of localized surface plasmon. *Journal of Physical Chemistry Letters* **1**, 2327–2333 (2010).
15. Shoji, T. *et al.* Permanent Fixing or Reversible Trapping and Release of DNA Micropatterns on a Gold Nanostructure Using Continuous-Wave or Femtosecond-Pulsed Near-Infrared Laser Light. *J Am Chem Soc* **135**, 6643–6648 (2013).
16. Righini, M. *et al.* Nano-optical Trapping of Rayleigh Particles and *Escherichia coli* Bacteria with Resonant Optical Antennas. *Nano Lett* **9**, 3387–3391 (2009).
17. Jones, S., Al Balushi, A. A. & Gordon, R. Raman spectroscopy of single nanoparticles in a double-nanohole optical tweezer system. *Journal of Optics* **17**, 102001 (2015).
18. Erickson, H. P.. *Biol Proced Online* **11**, 32–51 (2009).
19. Chen, Q. & Yuan, Y. J. A review of polystyrene bead manipulation by dielectrophoresis. *RSC Adv* **9**, 4963–4981 (2019).

REVIEWERS' COMMENTS

Reviewer #1 (Remarks to the Author):

The authors have suitably addressed all but one of my comments (regarding not using fluorescence to generalize the technique); however, as they point out, this is not the purpose of the manuscript, and the new experiments on speed up at different concentrations make the results even more compelling. Therefore, I strongly recommend publication.

Reviewer #2 (Remarks to the Author):

The authors have done a good job of addressing all of my comments from my previous review. I have no further suggested revisions and recommend for publication.